# The fate of land evaporation – A global dataset

Andreas Link[1], Ruud van der Ent[2,3], Markus Berger[1], Stephanie Eisner[4], Matthias Finkbeiner[1]

[1]Chair of Sustainable Engineering, Technical University of Berlin, Berlin, 10623, Germany
[2]Department of Water Management, Faculty of Civil Engineering and Geosciences, Delft University of Technology, Delft, Netherlands
[3]Department of Physical Geography, Faculty of Geosciences, Utrecht University, Utrecht, Netherlands
[4]Norwegian Institute of Bioeconomy Research, Ås, 1431, Norway

*Correspondence to*: Andreas Link (andreas.link@tu-berlin.de)

**Abstract.** Various studies investigated the fate of evaporation and the origin of precipitation. The more recent studies among them were often carried out with the help of numerical moisture tracking. Many research questions could be answered within this context such as dependencies of atmospheric moisture transfers between different regions, impacts of land cover changes on the hydrological cycle, sustainability related questions as well as questions regarding the seasonal and inter-annual variability of precipitation. In order to facilitate future applications, global datasets on the fate of evaporation and the sources of precipitation are needed. Since most studies are on a regional level and focus more on the sources of precipitation, the goal of this study is to provide a readily available global dataset on the fate of evaporation for a fine-meshed grid of source and receptor cells. The dataset was created through a global run of the numerical moisture tracking model WAM-2layers and focused on the fate of land evaporation. The tracking was conducted on a $1.5° \times 1.5°$ grid and was based on reanalysis data from the ERA-Interim database. Climatic input data were incorporated in 3- to 6-hourly time steps and represent the time period from 2001 to 2018. Atmospheric moisture was tracked forward in time and the geographical borders of the model were located at +/- 79.5° latitude. As a result of the model run, the annual, the monthly as well as the inter-annual average fate of evaporation was determined for 8684 land grid cells (all land cells except those located within Greenland and Antarctica) and provided via source-receptor matrices. The gained dataset was complemented via an aggregation to country and basin scales in order to highlight possible usages for areas of interest larger than grid cells. This resulted in data for 265 countries and 8223 basins. Finally, five types of source-receptor matrices for average moisture transfers were chosen to build the core of the dataset: land grid cell to grid cell, country to grid cell, basin to grid cell, country to country, basin to basin. The dataset is, to our knowledge, the first ready-to-download dataset providing the overall fate of evaporation for land cells of a global fine-meshed grid in monthly resolution. At the same time, information on the sources of precipitation can be extracted from it. It could be used for investigations into average annual, seasonal and inter-annual sink and source regions of atmospheric moisture from land masses for most of the regions in the world and shows various application possibilities for studying interactions between people and water such as land cover changes or human water consumption patterns. The dataset is accessible under https://doi.pangaea.de/10.1594/PANGAEA.908705 (Link et al., 2019a) and comes along with example scripts for reading and plotting the data.

**1) Introduction**

Where does evaporated water go to and where is the origin of precipitation? This question has been addressed by more and
more studies within the last decades as demonstrated in more detail below. In order to describe the fate of evaporation or the
source of precipitation, the concept of atmospheric watersheds was developed in which the terms "evaporationshed" (Van der
Ent and Savenije, 2013) and "precipitationshed" (Keys et al., 2012) were introduced. According to van der Ent (2014), "an
evaporationshed describes the downwind atmosphere and surface that receives precipitation from a specific location's
evaporation", whereas "a precipitationshed is defined as the upwind atmosphere and surface that contributes evaporation to a
specific location's precipitation".

Several methods are available to identify the origin and fate of moisture such as analytic box models as well as physical and
numerical (Eulerian and Lagrangian) moisture tracking models (Gimeno et al., 2012). Particularly relevant for large scale
studies are numerical moisture tracking models, which were used in the majority of the more recent studies within this field
(Dominguez et al., 2019; Van der Ent et al., 2013; Gimeno et al., 2012). Those models show various application opportunities
of which some of the main applications are listed and partly exemplified below:

1.   Gaining increased knowledge on how regions of interest are dependent on the moisture supply from other regions
     (Bagley et al., 2012; Dirmeyer et al., 2009; Dominguez et al., 2016; Guo et al., 2019; Keune and Miralles, 2019; Keys
50        et al., 2012, 2018; Salih et al., 2016; Staal et al., 2018; Zhao et al., 2016, 2019)
2.   Understanding land cover changes and their impacts on the supply of moisture to downwind beneficiaries (Bagley et
     al., 2012; Keys et al., 2012, 2018; Spracklen et al., 2012; Staal et al., 2018; Tuinenburg et al., 2012; Wang-Erlandsson
     et al., 2018; Wei et al., 2013, 2016)
3.   Applications within the context of sustainability and Water Footprinting (Berger et al., 2014, 2018)
4.   Understanding the seasonality of precipitation (Guo et al., 2019; Miralles et al., 2016; Zhang et al., 2017) as well as
     its inter-annual variability (Guo et al., 2019; Keys et al., 2018; Sodemann et al., 2008)
5.   Understanding precipitation changes and trends (Zhang et al., 2017, 2019)
6.   Investigations into impacts of climate change on the hydrological cycle (Bosilovich et al., 2005; Findell et al., 2019;
     Singh et al., 2016, 2017)
7.   Understanding extreme weather events such as droughts and floods (Dirmeyer and Brubaker, 1999; Drumond et al.,
     2019; Gangoiti et al., 2011; Gimeno et al., 2016; Herrera-Estrada et al., 2019; Nieto et al., 2019)

The first application refers to moisture supply dependencies for specific regions of interest and comes practically often along
with questions related to land cover changes. It can be of importance for regions which mainly rely on rain-fed agriculture
where changes in local precipitation could lead very likely to effects on agricultural yields (Van der Ent, 2014; Rockström et

al., 2009). Bagley et al. (2012) used in this regard results of a numerical moisture tracking in order to gain knowledge about the sources of precipitation for the major food producing regions in the world. They analyzed the vulnerability of regions towards a decline in crop productivity while including simulations of alterations in land covers of surrounding regions (Bagley et al., 2012). Besides regions of rain-fed agriculture, further regions of interest in research are rainforests or urban areas. Staal

et al. (2018) investigated, for instance, cascading moisture recycling effects of the Amazon rainforest, whereas Keys et al. (2018) determined the sources of precipitation and water security challenges for various megacities. Next to investigations into moisture supply dependencies and land cover changes, methods and tools within the context of sustainability are listed as a further potential application possibility. One method which could be named in this context is the Water Footprinting, which quantifies the water consumption as well as the resulting potential environmental impacts along a product's life cycle

(International Organization for Standardization, 2016). First considerations to include moisture tracking in Water Footprinting have been accomplished by Berger et al. (2014, 2018). The last application focus exemplified here refers to a deeper understanding on seasonality aspects as well as the inter-annual variability of precipitation. Guo et al. (2019), for instance, investigated within this context the moisture sources for East Asian precipitation and their temporal variability.

In order to facilitate future applications with regard to atmospheric watersheds, global datasets on the fate of evaporation and the sources of precipitation are needed. However, to our knowledge, only one large-scale approach exists so far which tried to track atmospheric moisture globally over a fine-meshed grid: Dirmeyer et al. (2009) used Lagrangian numerical moisture tracking to determine the sources of precipitation for all land cells across a $1.9° \times 1.9°$ grid. This resulted in an estimation of the source regions of precipitation for most nations as well as major basins in the world made publicly available online (DelSole

and Dirmeyer, 2012; Dirmeyer et al., 2009).

A comprehensive and global dataset on the fate of land evaporation was so far not readily available to the broader scientific community. Therefore, the goal of this study is to develop a global scale dataset on the fate of land evaporation for a fine-meshed grid of source and receptor cells which is openly available in a long-term data repository. The results of the study will

be presented as source-receptor matrices depicting the yearly average moisture transfers between grid cells. Besides yearly averages, the dataset will comprise monthly averages as well as data in inter-annual resolution. The dataset should enable researchers to gain comprehensive information on the fate of evaporation for any land area of interest covered by the model. Additionally, the goal is to provide information about source-receptor matrices for land areas of a high potential interest such as countries or basins.

**2) Material and methods**

We used the Eulerian numerical moisture tracking model WAM-2layers (Water Accounting Model-2layers) to create the dataset, which is able to spatially track tagged moisture forward and backward in time – on regional as well as on global scales

(Van der Ent, 2014). The WAM-2layers method as well as its predecessor version has been used extensively (e.g. in Van der Ent and Savenije, 2013; Findell et al., 2019; Guo et al., 2019; Keys et al., 2012, 2018; Keys and Wang-Erlandsson, 2018;

Wang-Erlandsson et al., 2018; Zemp et al., 2017; Zhang et al., 2017, 2019; Zhao et al., 2016) and showed results which were consistent with studies using other tracking methods (Van der Ent et al., 2013). We applied the Python version of the model, which is available on GitHub (Van der Ent, 2019), and modified pre- and post-processing. The atmospheric moisture tracking was conducted forward in time, thus focusing on the fate of evaporation. The considered grid covered the globe from 79.5° N to 79.5° S latitude. Calculations were performed on a 1.5° latitude × 1.5° longitude grid leading to a total amount of 25680

grid cells (107 × 240). In order to reduce the computational costs, the amount of cells for which the tracking has been applied was reduced to cells which contain land masses or are located within bigger inland lakes (e.g. the Caspian Sea). The land masses of Greenland and Antarctica were excluded because Eulerian moisture tracking at high latitudes is prone to errors due to high wind speeds compared to the size of the grid cell. As a result, 8684 cells were targeted for the atmospheric moisture tracking. The exact geographical information on the grid and the cells considered for tracking were summarized and are part

of the provided dataset.

ERA-Interim (ERA-I) reanalysis data were used as input for the model, which are provided by the European Centre for Medium Range Weather Forecasting (ECMWF) (Berrisford et al., 2011; Dee et al., 2011). The considered time horizon for the input data refers to the period of 2000 to 2018. However, the results are going to be presented for the period of 2001 to 2018 as the

first year was used as a model spin-up. The following data items were used as input parameters for the model:

- o Evaporation, precipitation
- o Wind components in zonal and meridional direction
- o Specific humidity
o Surface pressure
- o Total column water, total column water vapor
- o Vertical integral of eastward water vapor flux, vertical integral of eastward cloud liquid water flux, vertical integral of eastward cloud frozen water flux
- o Vertical integral of northward water vapor flux, vertical integral of northward cloud liquid water flux, vertical integral
of northward cloud frozen water flux

Evaporation and precipitation inputs were incorporated on a three-hourly basis. All other data items were integrated into the model on a six-hourly basis. The download of the data occurred at model levels spanning the atmosphere from zero pressure to surface pressure, which are broken down by the model to two layers with well-mixed conditions. The point of division

depends on the surface pressure (Van der Ent et al., 2014, Eq. (B5)), but is at approximately 2 km height for a standard surface pressure of 101325 Pa. This division was found to best represent sheared wind systems with wind in the bottom layer going to

another direction than wind in the top layer and is most relevant within the tropics where wind shears are particularly strong and a single layer assumption would be too fault-prone (Van der Ent et al., 2013, Figure 11; Goessling and Reick, 2013, Figure 3)


The underlying principle of the WAM-2layers model is the water balance shown in Eq. (1), which was applied in a replicate manner for each time step across the entire grid:

$$\frac{\partial S_k}{\partial t} + \frac{\partial (S_k u)}{\partial x} + \frac{\partial (S_k v)}{\partial y} = E_k - P_k + \xi_k \pm F_v \,, \tag{1}$$

$S_k$ represents the atmospheric moisture storage in layer $k$ and $t$ stands for time. The subscript $k$ stands either for the top or the

bottom layer. The variables $u$ and $v$ are describing the wind directions in zonal ($x$) and meridional ($y$) directions and represent the horizontal moisture transport between grid cells. Evaporation entering a layer is described by $E_k$ and precipitation removed from a layer by $P_k$. $\xi_k$ is a residual, which is a result of data-assimilation in ERA-I as well as of different spatial and temporal resolutions in the calculation steps of the WAM-2layers model. The last term of the equation ($F_v$) describes the vertical moisture transport between the two layers. This term is the one most difficult to calculate due to dispersive moisture exchange

besides transport by average vertical wind speeds (Dominguez et al., 2019). In WAM-2layers it is assumed it to be the closure term of the water balance. However, complete closure is not always possible and the net vertical fluxed was determined in such as that the water balance error is moisture-weighted equal for both layers. The gross vertical flux is parameterized to be 4 times the net flux in the direction of the net flux and 3 times the net flux in the opposite direction. More detailed information on the determination of all single terms from Eq. (1) is given in the work of van der Ent et al. (2014, appendix B)


The main calculations were conducted on the massively parallel computing system of the North-German Supercomputing Alliance (HLRN). During a first post-processing, the results were then aggregated to 13 source-receptor matrices with 8684 × 25680 cells – twelve for the monthly averages and one for the yearly average moisture transfers of the considered time period. Besides the yearly and monthly averages, matrices were also compiled on an inter-annual basis. Table 1 exemplifies the general

structure of a source-receptor matrix. The source cells refer within this context to land cells only, whereas the receptor cells cover the whole considered grid.

**Table 1 Exemplary source-receptor (evaporation-precipitation) matrix – source cells refer to considered land cells only, whereas receptor cells cover all grid cells between 79.5° N and 79.5° S latitude**

| Source-receptor matrix | Source cell 1 | Source cell 2 | … | Source cell 8684 |
|---|---|---|---|---|
| Receptor cell 1 | … | … | … | … |
| Receptor cell 2 | … | … | … | … |
| … | … | … | … | … |
| Receptor cell 25680 | … | … | … | … |

According to Eq. (2), we verified in each case how well the water balance closes:

$$\Delta_{\text{closure}} = (E_{\text{input}} - E_{\text{assigned}} - L_{\text{north}} - L_{\text{south}} - L_{\text{system}}) / (E_{\text{input}}) \tag{2}$$

where $\Delta_{closure}$ represents the mismatch within the water balance, $E_{\text{input}}$ is the amount of evaporation input, $E_{\text{assigned}}$ is water tracked over the considered grid until the point of re-precipitation, $L_{\text{north}}$ and $L_{\text{south}}$ are unassigned fractions of tracked water which got lost via the system boundaries (latitudes higher than 79.5° N/S) and $L_{\text{system}}$ are system losses. The latter term describes

unassigned water that is 'lost' from the system in the rare case the tracked water would exceed the total water. It may occur especially over mountainous areas or during heavy rainfall whereby the simplified offline tracking does not correspond to the more advanced weather model of ERA-I, or it may be caused by imbalances due to data-assimilation in ERA-I. Mismatches within the water balance could occur because we tracked moisture for all months simultaneously while using the simplified assumption that the water supply from month N-x to month N will approximately be the same as from month N to month N+x.

However, the reality might certainly be characterized in addition by cross-period moisture transfers.

    In order to also develop source-receptor matrices for larger regions of interest, moisture transfers of grid cells located within basins or countries were aggregated. Grid cells which contributed only partly to a basin or country were allocated according to the extent of overlap with the respective target area. The described procedure was done with the help of the ArcGis software

in which firstly a country and secondly a basin layer was overlain with the 1.5° × 1.5° grid. With regard to countries, the global country boundaries from DIVA-GIS with 265 countries were used, which were provided on the ArcGis website (Cun, 2016). We highlight that we do not have any political intentions by referring to this list and that we used it merely as a means of exemplification. Regarding the basins, the basin mask from the Watergap3 model (Eisner, 2016) was applied for the overlaying. Due to geographical boundaries at 79.5° latitude N / S, 8223 basins were considered in total. After the overlaying

of the respective maps, the geometric intersections were determined within ArcGis. This was followed by a post-processing in Python dedicated to the creation of the final source-receptor matrices for countries and basins. Finally, the following five types of source-receptor matrices for average moisture transfers were chosen to build the core of the dataset: land grid cell to grid cell, country to grid cell, basin to grid cell, country to country, basin to basin. With regard to the latter two matrices, the quantification of moisture contributions to and from the sea was targeted in addition to moisture transfers between countries

and basins, respectively. This was achieved as follows:

o    A country's or basin's share of precipitation originating from the sea was calculated via the difference in total precipitation and the sum of precipitated water originating from countries (or basins)

o    A country's or basin's total amount of evaporated water which re-precipitates over the sea was calculated via the difference

between the re-precipitation over the whole grid and the one taking place over the sum of countries (or the sum of basins).

Finally, usage possibilities of the created dataset were shown via site-specific examples. Examples were chosen with the objective to cover at least all continents and a wide variety of climate zones.

## 3) Results

### 3.1) Source-receptor matrices

The gained source-receptor matrices represent the main results of the created dataset. Table 2 specifies the different matrix types, the allocation of source and receptor regions to columns and rows as well as the numbers of matrices.

**Table 2 Source-receptor matrices of the created dataset (type 1: land grid cell to grid cell, type 2: country to grid cell, type 3: basin to grid cell, type 4: country to country, type 5: basin to basin)**

| Type | From (source) – Matrix columns | To (receptor) – Matrix rows | Number of matrices |
|---|---|---|---|
| 1 | 8684 land grid cells | 25680 grid cells | 13 (monthly + yearly averages) + 18 * 13 (separate inter-annual data for the years 2001 to 2018) |
| 2 | 265 countries | 25680 grid cells | 13 (monthly + yearly averages) |
| 3 | 8223 basins | 25680 grid cells | 13 (monthly + yearly averages) |
| 4 | 265 countries + sea | 265 countries + sea + unassigned | 13 (monthly + yearly averages) |
| 5 | 8223 basins + sea | 8223 basins + sea + unassigned | 13 (monthly + yearly averages) |

Particularly important is the provision of type 1 matrices within the dataset as they represent the raw data on a grid cell basis from which any further aggregation to larger land areas of interest could potentially take place. Together with the type 2 (country to grid) and type 3 (basin to grid) matrices, they enable the plotting of evaporationsheds over the whole area of the considered grid. The matrices of type 4 and 5, on the other hand, allow for the generation of self-explanatory source-receptor tables between countries and basins, respectively.

Besides the relevant source-receptor matrices, mismatches within the water balance ($\Delta_{closure}$) as well as all other terms of Eq. (2) are provided within the dataset. Identified mismatches are in general negligibly small for the annual averages (on average 0.03 % for land grid cells) but reach higher values on a monthly basis (on average 12.6 % for land grid cells). Unassigned fractions of moisture were exclusively allocated to losses via the northern and southern boundaries of the model. Thus, system losses due to storage limits play no role at all.

### 3.2) Visualization of sample evaporationsheds

Figure 1 to Figure 3 display the yearly average evaporationsheds for three chosen land grid cells, countries and basins. Based on sample scripts provided within the dataset, these types of figures can be plotted for any land grid cell, country or basin of interest. An additional online viewer can be used to directly look up the plots for any land grid cell. Re-precipitation of

evaporated water takes place over the whole considered grid and is expressed as a percentage of the evaporated water from the source region. The threshold for the plotting of re-precipitation within different grid cells lies at 0.02 % from the total amount of the assigned water. Additional information with regard to the location, the total evaporation input into the system ($E_{input}$), the unassigned fractions of water as well as the total share of re-precipitation displayed via the plot are provided seperately via

the image captions. Monthly information on moisture transfers for the chosen examples are available within the supporting information (Figure S1 to Figure S36).

Figure 1 shows the evaporationsheds for land grid cells located at Kansas City, US (a), Delhi, India (b) and Kampala, Uganda (c). It exemplifies different possible shapes and geographical extents of evaporationsheds. The evaporationshed for the source

cell at Kansas City sprawls, for instance, over large distances and does still not cover more than 70.0 % of the assigned re-precipitation. The evaporationsheds for the source cells at Kampala and Delhi, on the other hand, cover considerably higher shares of the assigned re-precipitation (79.0 % (b), 88.8 % (c)). With regard to the source cell at Kampala, huge amounts of moisture re-precipitate close to the source of evaporation and, thereof, more than 5 % within the source cell itself. Re-precipitation of evaporated water occurs here mainly westwards from the source cell along the equatorial belt and covers huge

areas of Central Africa. For the other source cells, moisture recycling takes place mainly eastwards (Kansas City) and southeastwards (Delhi) with lower shares of re-precipitation close to the source of evaporation. The tracking of atmospheric moisture for the source cell at Kansas City led to slight boundary losses due to the loation near to the northern boundary of the model. With regard to Delhi and Kampala, unassigned fractions of moisture due to losses of tagged moisture via the northern or southern boundaries are negligible.


Figure 2 displays evaporationsheds for the example countries Brazil (a), Egypt (b) and Laos (c). Brazil shows a non-fragemented evaporationshed with a huge amount of moisture recycling occuring within the country itself. Egypt's evaporationshed is fragmented with moisture recycling taking place close to the equatorial belt, over the Mediterranean and in the southeast of Europe and Asia. However, hardly any re-precipitation occurs within the country. The evaporationshed of

Laos is again non-fragemented with main areas of moisture recycling in Southeast Asia, over the sourrounding sea or in China.

Figure 3 finally presents the example evaporationsheds for basins referring to parts of the Rio Grande (a), the Danube (b) and the Murray-Darling (c) basin. Core areas of moisture recycling are Central and North America (a), the equatorial belt and huge parts of Eurasia (b), Northern and Eastern Autralia as well as the South Pacific Ocean (c). Displayed evaporationsheds are

large while covering only 59.4 to 70.1 % of the assigned moisture recycling.

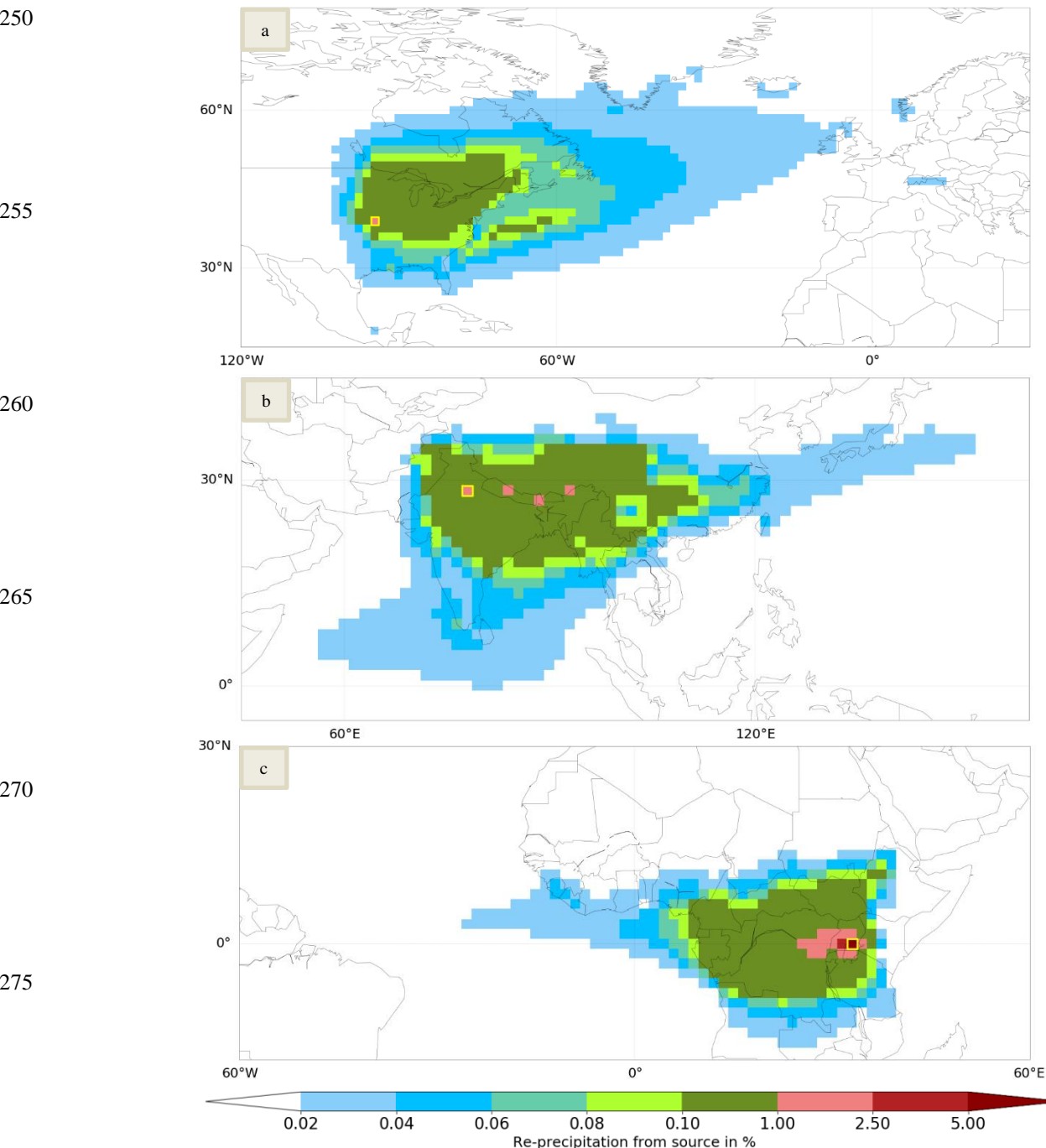

**Figure 1 Examples for yearly evaporationsheds of grid cells: a) cell at 39.0° N latitude & 94.5° W longitude (Kansas City, US), $E_{input}$ : 871.6 mm/a,  Unassigned : 2.3 %, Colored area covers 70.0 % of the assigned water b) cell at 28.5° N latitude & 78.0° E longitude (Delhi, India), $E_{input}$ : 1132.7 mm/a,  Unassigned : 0.1 %, Colored area covers 79.0 % of the assigned water c) cell at 0.0° latitude & 33.0° E longitude (Kampala, Uganda), $E_{input}$ : 1145.1 mm/a,  Unassigned : 0.0 %, Colored area covers 88.8 % of the assigned water**

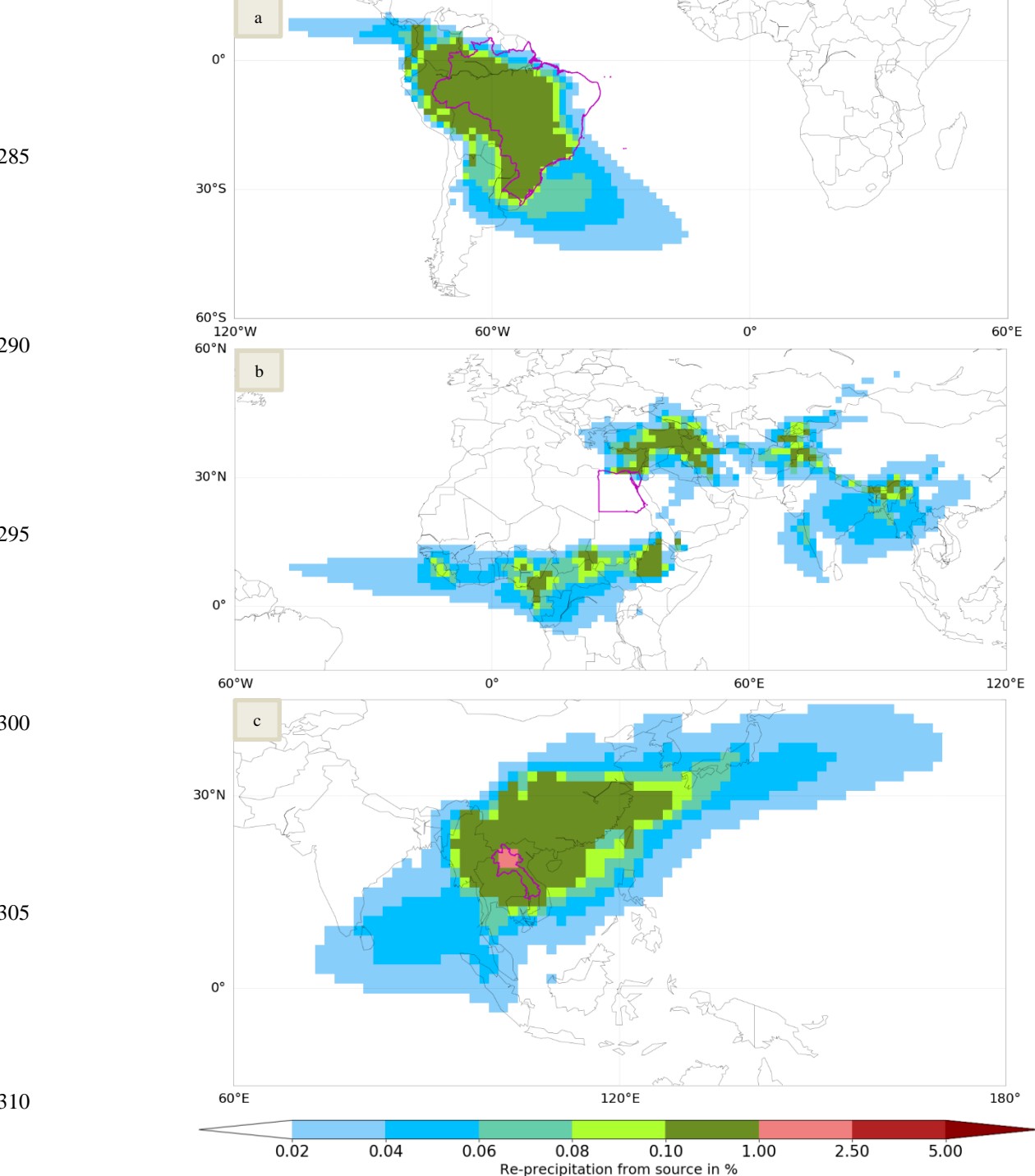

**Figure 2 Examples for yearly evaporationsheds of countries a) Brazil – $E_{input}$ : 1240.2 mm/a, Unassigned : 0.1 %, Colored area covers 80.4 % of the assigned water b) Egypt – $E_{input}$ : 104.0 mm/a, Unassigned : 0.8 %, Colored area covers 59.9 % of the assigned water c) Laos – $E_{input}$ : 1178.9 mm/a, Unassigned : 0.4 %, Colored area covers 77.9 % of the assigned water**







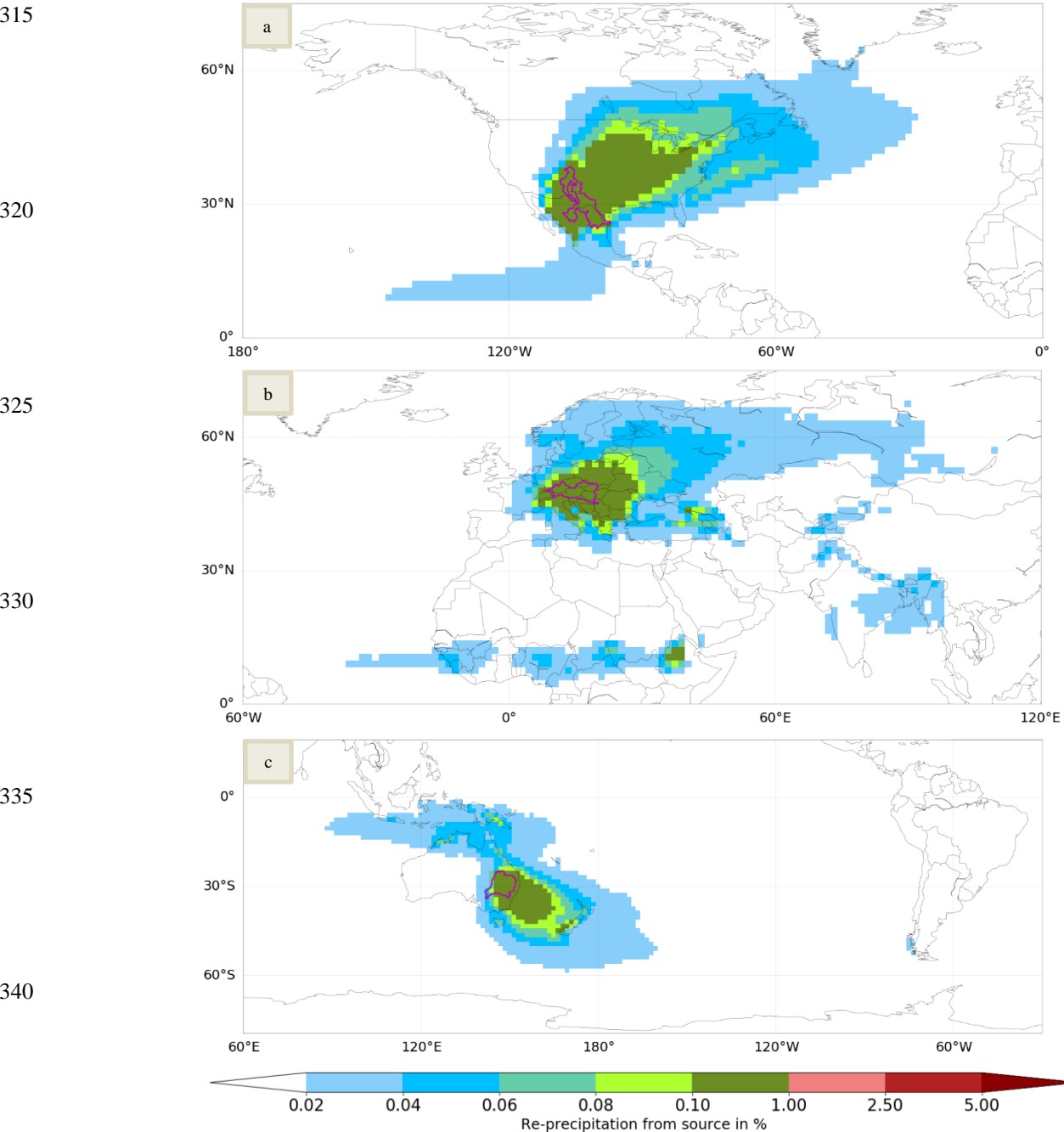

**Figure 3 Examples for yearly evaporationsheds of basins a) Basin ID 1463188 (part of the Rio Grande basin) - *E*input : 502.4 mm/a, Unassigned : 1.3 %, Colored area covers 70.1 % of the assigned water b) Basin ID 1019324 (part of the Danube basin) - *E*input : 609.4 mm/a, Unassigned : 4.0 %, Colored area covers 60.6 % of the assigned water c) Basin ID 2245569 (part of the Murray-Darling basin) - *E*input : 503.5 mm/a, Unassigned : 0.5 %, Colored area covers 59.4 % of the assigned water**

### 3.3) Examples for source-receptor tables

Besides the visualization of evaporationsheds, the dataset enables a direct quantification of average moisture transfers between countries or basins within source-receptor tables. This aspect refers to the latter two matrix types (type 4 and 5). At this point, type 4 matrices (countries) are used to demonstrate the usage of both types of matrices. Table 3 shows the fate of evaporated water as well as the sources of precipitation for the selected countries. For comparative purposes, the same countries are displayed as for the plotting examples in Figure 2. The presented information is in each case limited to the top 10 sites of re-precipitation and the top 10 sources of precipitation. Values are provided in percent and are related to the total amount of the evaporation or the precipitation input.

**Table 3 Fate of evaporation and source of precipitation – Examples for country tables**

| Brazil | | | | Egypt | | | | Laos | | | |
|---|---|---|---|---|---|---|---|---|---|---|---|
| Evaporation: 1240.2 mm/a | | | | Evaporation: 104.0 mm/a | | | | Evaporation: 1178.9 mm/a | | | |
| Precipitation: 1868.4 mm/a | | | | Precipitation: 13.1 mm/a | | | | Precipitation: 2176.9 mm/a | | | |
| Fate of evaporation | | Origin of precipitation | | Fate of evaporation | | Origin of precipitation | | Fate of evaporation | | Origin of precipitation | |
| Site | In % | Site | In % | Site | In % | Site | In % | Site | In % | Site | In % |
| Brazil | 43.6 | Sea | 63.3 | Sea | 31.3 | Sea | 76.6 | Sea | 44.5 | Sea | 70.0 |
| Sea | 33.6 | Brazil | 28.9 | Russia | 7.3 | Egypt | 2.7 | China | 26.1 | Thailand | 6.4 |
| Peru | 4.7 | Bolivia | 1.2 | China | 5.6 | Turkey | 1.9 | Laos | 7.5 | Laos | 4.1 |
| Colombia | 4.5 | Peru | 0.6 | India | 5.0 | Greece | 1.2 | Vietnam | 5.0 | India | 3.9 |
| Bolivia | 4.3 | Argentina | 0.6 | Ethiopia | 4.7 | Libya | 1.1 | Burma | 4.2 | Burma | 3.6 |
| Argentina | 4.0 | Angola | 0.4 | Iran | 3.5 | Sudan / South Sudan | 0.9 | Thailand | 4.2 | China | 3.4 |
| Paraguay | 1.5 | Paraguay | 0.4 | Sudan / South Sudan | 3.5 | Algeria | 0.9 | India | 1.2 | Vietnam | 1.9 |
| Ecuador | 1.2 | Venezuela | 0.3 | Turkey | 3.4 | Nigeria | 0.8 | Russia | 1.1 | Cambodia | 1.3 |
| Venezuela | 0.8 | Guyana | 0.3 | Kazakhstan | 2.3 | United States | 0.8 | Indonesia | 0.9 | Indonesia | 0.4 |
| Uruguay | 0.6 | Colombia | 0.3 | DR Congo | 2.2 | Italy | 0.8 | Cambodia | 0.8 | Russia | 0.3 |

The presented shares with regard to the fate of evaporation are in line with the visualization of evaporationsheds in Figure 2. For Brazil, the highest share of re-precipitation takes place within the country (43.6 %). With regard to Egypt and Laos, the highest share of evaporated water re-precipitates over the sea (Egypt: 31.3 %, Laos: 44.5 %). Concerning additional information on the origin of precipitation, Table 3 highlights the following: In all cases the sea is the biggest source of precipitation with values ranging from 63.3 % (Brazil) to 76.6 % (Egypt). With regard to Brazil and Egypt, the most important terrestrial source of precipitation is the country itself (Brazil: 28.9 %, Egypt: 2.7 %). The most relevant terrestrial evaporative source for the precipitation in Laos is Thailand, which supplies on average 6.4 % of the local precipitation.

## 4) Discussion

### 4.1) Possible uses of the dataset

The introduction already provided a broad overview on various uses of numerical moisture tracking. In the following it will be summarized for which of the named applications our created dataset could be particularly suitable. The first presented application referred to an increased knowledge on how regions of interest are dependent on the moisture supply from other regions. The provided dataset could provide valuable information to answer those questions but shows the following limitation: While the dataset includes comprehensive information on the fate of evaporation, information regarding the sources of precipitation are limited to land areas and cannot displayed across the whole grid of land and sea cells. The reason for this is the chosen tracking direction (forward in time) and the focus on land grid cells for the tracking in order to reduce to computational efforts. Nevertheless, the dataset quantifies the amount of precipitation originating from the sea without knowing the exact non-terrestrial source locations. Examples for this were given in Table 3.

The second presented application was related to predictions of potential impacts of human-induced land cover changes on the water cycle. The created dataset could serve as an estimate for the question how land cover changes and altered amounts of land evaporation would potentially affect the supply of water via re-precipitation elsewhere (Keys et al., 2012). Van der Ent et al. (2010) stated within this context that decreasing evaporation (e.g. via deforestation) for areas with high shares of moisture recycling over land "would enhance droughts in downwind areas where overall precipitation amounts are low". The opposing statement to that would also be conceivable – namely that increased land evaporation in these areas could also result in positive water supply effects. Such first-order estimates are relevant in the context of socio-hydrology (Keys and Wang-Erlandsson, 2018; Sivapalan et al., 2012), but we highlight at this point that the dataset can generally not provide more than rough estimates regarding this topic. An exception could be the inspection of inter-annual data for sites where major land cover changes occurred within the covered time period. However, for more comprehensive information on this subject it is advised to apply atmospheric moisture tracking directly to different land cover scenarios.

With regard to the third stated application, sustainability studies and Water Footprinting, the provided dataset shows as well promising usage possibilities. Knowledge on the fate of evaporation was firstly integrated within the method of Water Footprinting by Berger et al. (2014, 2018) via an enhanced water accounting method. This considered atmospheric moisture recycling ratios within drainage basins, which could reduce water consumption patterns significantly (Berger et al., 2014, 2018). Aspects of moisture recycling across basin boundaries have not yet been considered so far. The comprehensive information on the fate of land evaporation of the dataset could be used for research regarding this topic.

The fourth possible application was related to research on the variability of precipitation and included seasonal and inter-annual variabilities. As the dataset provides both – monthly data averaged over the considered time period as well as inter-

annual data – it shows a high suitability for this kind of usage. Limitations with regard to the usage of seasonal data could be related to possible mismatches in the water balance which should be verified before usage. However, for the yearly averages
those mismatches get negligibly small. The application of studying inter-annual variability, on the other hand, is limited to the covered time period (years 2001 to 2018).

Precipitation changes and trends represented the fifth application focus. The dataset can be used in this context to understand changes and trends of moisture recycling for the considered time period, whereas predictions into the future are not possible.
The sixth and seventh application were related to impacts of climate change on the hydrological cycle and the understanding of extreme weather events. The usage of the dataset for the determination of impacts related to climate change is limited to changes in climate which are reflected by the reanalysis data considered for this study. However, for a deeper analysis of the relationship between global temperature increases and resulting changes in moisture supply patterns, models including scenario analyses would be more suitable. With regard to the understanding of extreme weather events, the dataset could be used in
order to gain an increased knowledge on the causes for past droughts. This could be achieved via investigations into anomalies of moisture supply patterns for relevant locations and time periods covered by the model. Investigations into extreme weather events such as floods, on the other hand, are not possible with this dataset as those would require a modeling with higher spatial and temporal resolutions.

## 4.2) Critical reflections on the used input data

The following section deals with the critical reflection on the ERA-I data (Berrisford et al., 2011; Dee et al., 2011), which were used as input for the creation of the dataset. ERA-I, which has been updated during the process of the preparation of this article to ERA5 (Hersbach et al., 2020), "showed both a comparatively reasonable closure of the terrestrial and atmospheric water balance as well as a reasonable agreement with observation datasets" (Lorenz and Kunstmann, 2012). It has been frequently used to study the hydrological cycle (Li et al., 2019) and ranks among the best representations of the hydrological processes
within the atmosphere (Gao et al., 2014; Lorenz and Kunstmann, 2012). However, within the past also some biases were reported, especially with regard to the variables evaporation and precipitation (Bumke, 2016; Fu et al., 2016). Plots for these two variables are presented as daily averages in Figure S37 of the supporting information. Moreover, we provide in Figure S38 of the supplement a grid cell based comparison between ERA-I and its successor version ERA5. This revealed that the variations in evaporation (Figure S38, part a) and precipitation (Figure S38, part b) between the two data sources are relatively
small in most regions (<< 1mm). The differences in precipitation (Figure S38, part b), however, can take for a few connected regions as well higher values of up to 2, 3 or even more than 4 mm per day. Those can mainly be found within the high precipitation areas of the tropics and along the west coast of North and South America. Considering that ERA5 claims in particular an improved performance over land in the deep tropics (ECMWF, 2020; Hersbach et al., 2020), precipitation in ERA-I might for some of the tropical regions be slightly over- (e.g. in Central Africa) or underestimated (e.g. on Borneo).

Next to the grid cell based comparison of ERA-I to ERA5, we provide an additional analysis on continental scales. This compares the average continental evaporation and precipitation of ERA-I to ERA5 as well as a study by Rodell et al. (2015). The latter combined a variety of data sources such as GPCP v2.2 (Adler et al., 2003), SeaFlux v1.0 (Clayson et al., 2012), MERRA (Bosilovich et al., 2011), MERRA-Land (Reichle, 2012) and GLDAS (Rodell et al., 2004) to derive an observed state of the water cycle in the early 21$^{st}$ century. Methodological details regarding the comparison can be reviewed in the supporting

information. Table 4 presents the derived results, which cover all continents except Antarctica plus the overall world land, world ocean and earth as a whole. We stress that a final conclusion on which dataset is closest to reality is regarded as out of scope for this paper. We can, however, conclude that repeating our analysis with ERA5 would overall not lead to major differences. This is due to the fact that both the continental comparison (Table 4) as well as the grid cell based comparison (Figure S38) between ERA-I and ERA5 revealed for most regions generally high similarities. The comparison to Rodell et al.,

on the other hand, led to more significant differences. Table 4 demonstrates that the intensity of evaporation over land in both ERA-I and ERA5 seems overestimated compared to Rodell et al. (2015), especially in Australia (up to + 52.7 %) and Eurasia (up to + 21.6 %). A similar trend can be observed regarding the variable precipitation, where, except for North America and Australia, ERA-I and ERA5 show consistently higher values. With regard to precipitation over Australia, however, an opposing trend is visible. Here, ERA-I as well as ERA5 might possibly underestimate precipitation over land, which would be in line

with findings made by Fu et al. (2016) for this region.

**Table 4 Continental evaporation (E) and precipitation (P) of ERA-I (Berrisford et al., 2011; Dee et al., 2011) in comparison to ERA5 (Hersbach et al., 2020) and the study by Rodell et al. (2015)**

| Regions | Evaporation in mm per day | | | Δ in % | | Precipitation in mm per day | | | Δ in % | |
|---|---|---|---|---|---|---|---|---|---|---|
| | ERA-I | ERA5 | Rodell et al. (2015) | ERA-I to ERA5 | ERA-I to Rodell et al. (2015) | ERA-I | ERA5 | Rodell et al. (2015) | ERA-I to ERA5 | ERA-I to Rodell et al. (2015) |
| North America | 1.34 | 1.49 | 1.13 | - 10.1 % | + 18.6 % | 1.95 | 2.20 | 2.02 | - 11.4 % | - 3.5 % |
| South America | 3.00 | 2.97 | 2.67 | + 1.0 % | + 12.4 % | 4.96 | 5.40 | 4.57 | - 8.1 % | + 8.5 % |
| Eurasia | 1.41 | 1.40 | 1.16 | + 0.7 % | + 21.6 % | 2.09 | 2.18 | 1.98 | - 4.1 % | + 5.6 % |
| Africa | 1.76 | 1.75 | 1.54 | + 0.6 % | + 14.3 % | 2.13 | 1.92 | 1.89 | + 10.9 % | + 12.7 % |
| Oceania | 3.19 | 3.11 | 3.10 | + 2.6 % | + 2.9 % | 7.91 | 7.68 | 6.79 | + 3.0 % | + 16.5 % |
| Australia | 1.42 | 1.41 | 0.93 | + 0.7 % | + 52.7 % | 1.03 | 1.10 | 1.42 | - 6.4 % | - 27.5 % |
| World land | 1.59 | 1.61 | 1.32 | - 1.2 % | + 20.5 % | 2.31 | 2.40 | 2.18 | - 3.8 % | + 6.0 % |
| World ocean | 3.50 | 3.60 | 3.37 | - 2.8 % | + 3.9 % | 3.16 | 3.31 | 3.03 | - 4.5 % | + 4.3 % |
| World | 2.96 | 2.96 | 2.79 | +- 0 % | + 6.1 % | 2.91 | 3.05 | 2.79 | - 4.6 % | + 4.3 % |

Logically, at the end of this discussion, the question arises what users of the dataset could do if they find the ERA-I evaporation
or precipitation data unreliable while, at the same time, more representative data would be available. In this case, we recommend to solely use the relative source-receptor relationships of our dataset while plugging in own data regarding the absolute values of evaporation and precipitation. This assumption will likely be satisfactory in case all data is equally biased, but when only certain areas are considered biased a correction procedure would be more complicated.

### 4.3) Comparison to other data sets

At this point, a general comparison of our dataset to the existing one referring to the Lagrangian 3D quasi-isentropic back-trajectory (3D QIBT) method (DelSole and Dirmeyer, 2012; Dirmeyer et al., 2009) forced with the NCEP-DOE AMIP-II reanalysis (R-2) (Kanamitsu et al., 2002) and CMAP data (Xie and Arkin, 1997) is given. Next to a slightly higher spatial resolution (1.5° compared to 1.9° resolution), the results of our study are easier to access due to the publication of raw data and aggregated data in a public repository (ready-to-download data). An advantage of the dataset based on the 3D QIBT
method, on the other hand, is a longer considered time period (25 years to 18 years). A significant difference lies in the tracking direction of the two approaches. The 3D QIBT approach traces moisture generally backward in time and its application led to comprehensive information on the sources of precipitation. By contrast, our study focus was on analyzing the fate of evaporation, which was realized through a forward tracking of atmospheric moisture. The different tracking directions led to different opportunities for the plotting of atmospheric watersheds. Our dataset enables the plotting of evaporationsheds over
the whole considered grid of land and sea cells, whereas the plotting of precipitationsheds is limited to the areas of land. Vice versa, the dataset based on the QIBT method enables the plotting of precipitationsheds over the whole considered grid, whereas the plotting of evaporationsheds is limited to land cells. In order to exemplify differences of the study outputs, study results on a country level from Dirmeyer et al. (2009) were compared to the results of our dataset based on the following two data items:

o   Terrestrial evaporative source (TES = Fraction of precipitation that originated as evaporation from terrestrial sources) according to Dirmeyer et al. (2009)
       o   Country internal evaporative source (CIES = Fraction of precipitation that originated as evaporation from the same country), which is termed recycling ratio (RR) in Dirmeyer et al. (2009)

Table 5 and Table 6 analyze the top 10 countries with the highest and lowest average TES and CIES values for both datasets. As a general trend, our dataset shows in most cases a higher ocean contribution for the evaporative sources of precipitation (derived by in general lower TES values). The main reason for this is probably that the data used by Dirmeyer et al. show a land evaporation which equals on average almost the precipitation over land (ratio of land evaporation to land precipitation: 0.99) and thus allow hardly any runoff (Trenberth et al., 2011; Xie and Arkin, 1997). This fact leads inevitable to TES values
(as well as CIES values) which could be classified more on the high side. Moreover, there may be several methodological differences causing different output such as different (vertical) mixing assumptions.

**Table 5** Comparison of the top 10 countries with the highest and lowest average TES values between our dataset based on the WAM-2layers method and the one referring to the 3D QIBT method (Dirmeyer et al., 2009) - Countries appearing in both lists are displayed in bold font; CAR = Central African Republic

| Rank | Top 10 countries with the highest TES | | | | Top 10 countries with the lowest TES | | | |
|---|---|---|---|---|---|---|---|---|
| | WAM-2layers | in % | 3D QIBT | in % | WAM-2layers | in % | 3D QIBT | in % |
| 1 | **Mongolia** | 80.3 | **Mongolia** | 95.7 | **Chile** | 4.3 | **Chile** | 8.1 |
| 2 | Niger | 72.0 | **Paraguay** | 90.0 | **New Zealand** | 8.8 | **Portugal** | 9.9 |
| 3 | Chad | 68.0 | Nepal | 85.5 | **Philippines** | 9.3 | **New Zealand** | 9.9 |
| 4 | Mali | 66.8 | Namibia | 84.2 | **French Guiana** | 12.0 | Ireland | 11.1 |
| 5 | Cameroon | 64.0 | Bhutan | 84.0 | Papua New Guinea | 12.2 | **Philippines** | 11.6 |
| 6 | Burkina Faso | 63.0 | Russia | 83.2 | **Portugal** | 12.4 | Morocco | 12.7 |
| 7 | Mauritania | 62.8 | Botswana | 82.9 | Sri Lanka | 13.1 | Israel | 13.3 |
| 8 | **CAR** | 62.1 | Bolivia | 82.7 | Somalia | 14.5 | Lebanon | 13.7 |
| 9 | **Paraguay** | 61.9 | **CAR** | 82.0 | Suriname | 14.8 | **French Guiana** | 14.5 |
| 10 | Kyrgyzstan | 60.9 | Angola | 81.3 | Belize | 15.5 | United Kingdom | 14.9 |


**Table 6** Comparison of the top 10 countries with the highest and lowest average CIES values between our dataset based on the WAM-2layers method and the one referring to the 3D QIBT method (Dirmeyer et al., 2009) - Countries appearing in both lists are displayed in bold font; CAR = Central African Republic

| Rank | Top 10 countries with the highest CIES | | | | Top 10 countries with the lowest CIES | | | |
|---|---|---|---|---|---|---|---|---|
| | WAM-2layers | in % | 3D QIBT | in % | WAM-2layers | in % | 3D QIBT | in % |
| 1 | **Brazil** | 28.9 | **Russia** | 64.7 | **Luxembourg** | 0.2 | **Luxembourg** | 0.4 |
| 2 | **Russia** | 27.8 | Canada | 54.8 | **Qatar** | 0.3 | **Qatar** | 0.4 |
| 3 | **China** | 25.9 | **Brazil** | 46.3 | Lebanon | 0.5 | Belize | 0.5 |
| 4 | **DR Congo** | 25.1 | **United States** | 43.2 | **Gambia** | 0.8 | **Gambia** | 0.7 |
| 5 | Angola | 20.9 | **China** | 41.4 | **Israel** | 0.8 | **Israel** | 0.8 |
| 6 | **Australia** | 20.7 | **Australia** | 37.9 | Western Sahara | 0.9 | Equatorial Guinea | 1.2 |
| 7 | Argentina | 19.0 | **India** | 36.4 | Jordan | 0.9 | **Djibouti** | 1.3 |
| 8 | **United States** | 18.3 | Mongolia | 30.8 | **Djibouti** | 0.9 | El Salvador | 1.4 |
| 9 | **India** | 18.1 | **DR Congo** | 28.5 | Belgium | 1.0 | Macedonia | 1.4 |
| 10 | Sudan / South Sudan | 17.4 | Mexico | 28.4 | Iceland | 1.0 | Rwanda | 1.4 |

Next to general trends, different country compositions can be observed within the lists of the two datasets. Table 5 highlights that only three out of 10 countries appear for both datasets within the list of the 10 highest TES values (Mongolia, the Central African Republic (CAR) and Paraguay). In this context, Mongolia represents in each case the country with the highest share of precipitation originating from terrestrial sources (80.3 % in WAM-2layers, 95.7 % in 3D QIBT). Regarding the countries with the lowest TES values, both approaches list five countries in common (Chile, New Zealand, the Philippines, French

Guiana and Portugal) while showing the lowest value for Chile (4.3 % - WAM-2layers, 8.1 % - 3D QIBT). Regarding the CIES (Table 6), high values appear in general for relatively large countries. At this point, seven out of 10 countries are listed

for both datasets within the top 10 (Brazil, Russia, China, DR Congo, Australia, United States, India). The highest value refers to Brazil (28.9 %) for the WAM-2layers method and to Russia (64.7 %) for the 3D QIBT approach. Small CIES values, on the other hand, appear for relatively small countries. Here we find five countries in common (Luxembourg, Qatar, Gambia, Israel, Djibouti), with Luxembourg showing in each case the lowest value (0.2 % - WAM-2layers, 0.4 % - 3D QIBT). The fact that different countries appear in the tables is most likely caused by spatial differences of evaporation, precipitation and wind speed in the underlying reanalysis input data. Differences regarding the tracking method itself, on the other hand, might probably play a less important role as WAM-2layers was found to reach generally similar results to Lagrangian models (Van der Ent et al., 2013; Van der Ent and Tuinenburg, 2017). The overall comparison of the results for the TES and the CIES between the two methods including all countries can be gained from the supporting information (Table S2).

Larger overlaps between the two datasets could partly be identified while focusing on the top contributors for precipitation over individual countries. This is exemplified through Table S3 to S5 of the supporting information, which provide with regard to both datasets an overview on the top ten sources of precipitation for the sample countries Brazil, Egypt and Laos. Especially the country Laos shows in this context a relatively high match regarding the appearance of sources and their ranking to each other. A more detailed direct interpretation of the differences in the results between individual countries is at this point regarded as out of scope for this paper but could be tackled by comparative studies in the future.

**5) Data availability**

The dataset on the fate of land evaporation is available within the PANGAEA research data repository. It can be accessed through https://doi.pangaea.de/10.1594/PANGAEA.908705 and cited as Link et al. (2019a). The dataset consists of two sub datasets – a basic dataset which contains data averaged over the whole considered time period as well as an inter-annual dataset providing data for separate years. An attached PDF file ("readme.pdf") explains the structure of the dataset and gives all necessary information on how to work with it. In addition to the provided dataset, a screening tool for the visualization of evaporationsheds on a land grid cell to grid cell basis (based on matrix type 1 of Table 2) can be accessed through http://wf-tools.see.tu-berlin.de/wf-tools/evaporationshed/#/ (Link et al., 2019b).

**6) Conclusions**

The background of this research was an increased occurrence of studies on the fate and origin of atmospheric moisture. Numerical moisture tracking has been highlighted as one of the main methods to study those aspects. To our knowledge, so far only one approach had been published which tried to track atmospheric moisture globally over a fine-meshed grid (Dirmeyer et al., 2009). This aimed mainly at determining the sources of land precipitation (Dirmeyer et al., 2009). The goal of our study was the provision of a complementary publicly available high resolution global dataset on the fate of land

evaporation and was achieved via a global application of the numerical moisture tracking model WAM-2layers. A further post-processing resulted in monthly and yearly source-receptor matrices for average moisture transfers from land grid cells, countries and basins. Furthermore, raw data for inter-annual differences were compiled. The created dataset is the first publicly

available dataset ready-to-download providing the overall shape of evaporationsheds for land cells of a global fine-meshed grid in a monthly resolution. Additionally, information on precipitationsheds can be gained via the dataset. The dataset can be regarded as a useful complement to the existing dataset referring to the QIBT method (Dirmeyer et al., 2009; DelSole and Dirmeyer, 2012). It is expected that it will facilitate the access to data on atmospheric moisture recycling and could be integrated into future studies. Possible applications were identified and refer mainly to studies on atmospheric moisture

dependencies, impacts of land use changes, Water Footprinting, seasonality and inter-annual variabilities of precipitation, precipitation changes and trends as well as on droughts.

**Supplement.**

The supplement related to this article is available online at: https://essd.copernicus.org/preprints/essd-2019-246/essd-2019-246-supplement.pdf

**Author contributions.**

AL, RE: Adaption and testing of the python code, main model run, post-processing - AL, RE, MB: results presentation, plausibility checks, interpretation of the results and compilation of the dataset – AL, RE, MB, MF: Preparation of the manuscript – SE: Provision of the basin mask from the WaterGap3 model as well as advisory support for the post-processing in ArcGis.

**Competing interests.**

The authors declare that they have no conflict of interest.

**Acknowledgements.**

The authors acknowledge the HLRN for providing high performance computing resources that have contributed to the research results reported in this paper. Particularly, the support of Dr. Wolfgang Baumann from the HLRN concerning technical and

implementation aspects in making the code run on those resources is gratefully acknowledged. Furthermore, the authors acknowledge the support of the German Research Foundation (DFG) who founded this research within the project "Water Footprinting in the Manufacturing Industries – Methods, Tool and Optimization Strategies" (project number: FI 1622/4-1).

Ruud van der Ent acknowledges funding from the Netherlands Organization for Scientific Research (NWO), project number 016.Veni.181.015.

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

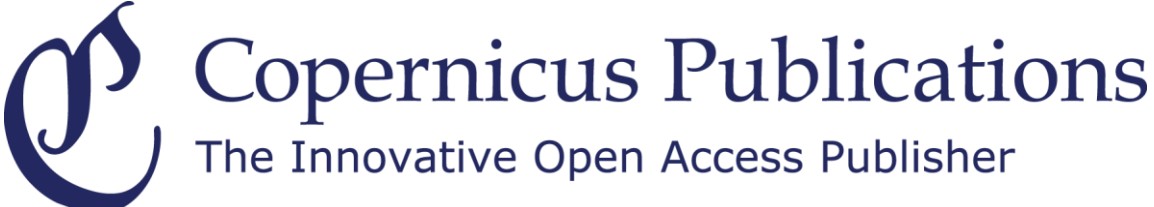

730   **Figure 4: The logo of Copernicus Publications.**