# Peer review of "The fate of land evaporation – A global dataset"

_Earth System Science Data, 2019_

## Referee Comment (RC1) · Anonymous Referee #1 · 16 Mar 2020

This manuscript presents a global dataset aiming to describe the moisture fate sourced from a particular area, ranging from a grid, a watershed to a country. Such dataset is not readily available and can be valuable for hydrometeorological studies trying to quantify the length scales and time scales of moisture transport starting from terrestrial evapotranspiration until it precipitates out and how these scales might change due to climate variability and land-cover land-use changes. The manuscript is well written and the authors did a great job in providing tools to analyze and visualize the data. However, because this dataset is the output from the WAM-2layers model to track moisture with a specific origin, I have two major concerns the authors should address before accepting for publication.

1. Because the WAM-2layer models divides the atmosphere into two layers, the paper

didn't provide adequate information regarding how the two layers are divided. It might be provided by the references, but I think it is important to explain this aspect in the "material and methods" section. In addition, please explain how the vertical moisture flux is obtained. Is the vertical wind at the interface of the two layers used or is it derived from water conservation at each layer? Please clarify. Also, the sensitivity of how these two layers is divided to the results, and how the 2-layer models improves the 1-layer results and where such improvements are the most evident should be fully discussed.

2. The presented dataset is based on one of the several methods/models that can be used to track water from evapotranspiration until it contributes to precipitation. I believe it is necessary to fully discuss the assumptions and limitations of this model, and anticipate how the results from WAM-2layer can be different from other models. Indeed, the authors have made such effort by comparing with QIBT estimates. However, it is important to discuss the possible biases, especially with respect to more sophisticated methods like water vapor tracers embedded in climate models, which tend to fully resolve the physical processes that moisture can possibly be involved in the climate models. Such discussion can be critical for the future users by providing caveats and advantages of this dataset and also how they can anticipate the differences with other methods and to be aware of when and where such differences will mostly likely occur.

---

## Referee Comment (RC2) · Anonymous Referee #2 · 26 Apr 2020

The manuscript "The fate of land evaporation - A global dataset" introduces a new dataset that can be used for computing exchanges of atmospheric moisture among different types of source-receptor pairs, including grid cells, major basins, and countries. This dataset comes from the output from a water tracking model, the WAM-2layers, and the ERA-Interim reanalysis. The authors present a list of potential applications for this new dataset, along with some simple examples of results, in the form of maps (of evaporationsheds) and tables (source-receptor matrices). The manuscript is well written and the presentation of the material is clear. The authors have done a good job both at producing the new dataset and at sumarizing the current general applications of water tracking studies and the particular potential applications of the new dataset. I think this manuscript can be improved to a final form by taking into acount the following

comments:

General comments:

1. The dataset was built using ERA-Interim data as input. Even when ERA-Interim had one of the best representations of different aspects of the hydrological cycle (now improved in ERA5), it also had some biases in representing variables like evaporation and precipitation. It would be helpful if the authors discuss the potential implications of these biases on the accuracy/applicability of their new dataset. For example, the authors could add maps of biases of evaporation and/or precipitation, at least for some regions. This would give the reader an idea about where the new dataset could have the largest uncertainties. A discussion about the biases in the input evaporation and precipitation could also help on the comparison with the QIBT estimates.

2. The current presentation of the comparison with estimates from the QIBT leaves the impression that there is little in common between both datasets, and no indication as to which dataset could be closer to a ground-truth. Given the accumulation of uncertainties, due to input reanalysis data and details of the water tracking methods, it is no surprise to have differences. However, not only there is kind of a systematic difference (where QIBT yields larger values than WAM-2layers, as discussed by the authors), but not even the rankings of the countries coincide between tables. Maybe, in order to look for information in both datasets that could be robust to the differences on input data and water tracking method, the authors could include an example of the ranking of sources for a given country, and check the consistency or lack of consistency (now in terms of ranks, not original fractions) between both datasets. This would provide the reader with a better sense on what information is the most robust in the new dataset.

Technical comments:

a. In Table 1, please add description of the "25680" and "8684" values, to help the reader to more easily understand the type of sources and receptors included in this table, without need to refer to distant parts of the paper (these specific values are

described well above (Lines 104-108), and then well below (Table 2)).

b. I tried the link for the visualization of the evaporationsheds (http://wftools.see.tu-berlin.de/wf-tools/evaporationshed/#/), but it did not work (on April 26, 2020). Please check.

c. Individual files (e.g. http://hs.pangaea.de/model/WAM-2layers/Link-etal_2019/Inter-annual/2018.zip) as in "LinkA-etal_2019_inter-annual.tab" are very large (19GB). It would be helpful to have smaller examples also available for download, for example for just one country or basin, in order to test the rest of tools available with this dataset (as in http://hs.pangaea.de/model/WAM-2layers/Link-etal_2019/readme.pdf). I think that having the possibility for this kind of simple tests would help the readers and encourage the potential users to actually download and work with this dataset.

---

## Author Comment (AC1) · 23 May 2020

The authors would like to thank both anonymous referees for their valuable feedback and provided improvement potential which is highly appreciated. In the following we will address the concerns and improvement potentials and highlight which changes we made within the manuscript.

A) Author's reply with regard to the comments of anonymous referee #1

A1.1) Comment 1

Because the WAM-2layer models divides the atmosphere into two layers, the paper didn't provide adequate information regarding how the two layers are divided. It might be provided by the references, but I think it is important to explain this aspect in the

"material and methods" section. In addition, please explain how the vertical moisture flux is obtained. Is the vertical wind at the interface of the two layers used or is it derived from water conservation at each layer? Please clarify. Also, the sensitivity of how these two layers is divided to the results, and how the 2-layer models improves the 1-layer results and where such improvements are the most evident should be fully discussed.

A1.2) Reply to comment 1

Thank you for this comment. We considered detailed information on how the layers were derived, how this improved the model and how the vertical moisture transport was determined as out of scope for this paper. Instead we referred to the following key references:

-Van der Ent, R. J.: A new view on the hydrological cycle over continents, Ph.D. thesis, Delft University of Technology, Delft, 96 pp., 2014.

-van der Ent, R. J., Wang-Erlandsson, L., Keys, P. W. and Savenije, H. H. G.: Contrasting roles of interception and transpiration in the hydrological cycle - Part 2: Moisture recycling, Earth Syst. Dyn., 5(2), 471–489, doi:10.5194/esd-5-471-2014, 2014.

-Van der Ent, R. J., Tuinenburg, O. A., Knoche, H. R., Kunstmann, H. and Savenije, H. H. G.: Should we use a simple or complex model for moisture recycling and atmospheric moisture tracking?, Hydrol. Earth Syst. Sci., 17(12), 4869–4884, doi:10.5194/hess-17-4869-2013, 2013.

However, we see the point that some descriptions could have been provied in a more detailed way. Thus, we provide within the manuscript now more information with regard to these points. Furthermore, we try to refer to the references in more detail, e.g. through giving the exact figure or appendix for the given references which provides in-depth insights on the respective sub-topics.

A1.3) Resulting changes within the manuscript

With regard to the comment on the 2 layers we inserted brief additional information

within the methods section on a) where the division of the layer can be found, b) how it was determined, c) and where this model improvement is of relevance. Furthermore, we refer now at two points to the references in more detail. These relate to the equation for the point of division for the two layers as well as to the occurrence of strong wind shears where a single layer assumption would be prone to errors. In the following, the made amendments will be presented:

->"The point of division depends on the surface pressure (Van der Ent et al., 2014, Eq. (B5)), but is at approximately 2 km height for a standard surface pressure of 101325 Pa. This division was found to best represent sheared wind systems with wind in the bottom layer going to another direction than wind in the top layer and is most relevant within the tropics where wind shears are particularly strong and a single layer assumption would be too fault-prone (Van der Ent et al., 2013, Figure 11; Goessling and Reick, 2013, Figure 3)"

With regard to the determination of the vertical moisture flux we added within the methods part a short section which gives the general idea of it without providing the exact equations.

->"The last term of the equation (Fv) describes the vertical moisture transport between the two layers. This term is the one most difficult to calculate due to dispersive moisture exchange besides transport by average vertical wind speeds. Thus, WAM-2layers assumed it to be the closure term of the water balance. However, complete closure is not always possible and the net vertical fluxed was determined in such as that the water balance error is moisture-weighted equal for both layers. The gross vertical flux is parameterized to be 4 times the net flux in the direction of the net flux and 3 times the net flux in the opposite direction."

More detailed information for the reader is accessible through the best suited reference, which is provided within the manuscript.

A2.1) Comment 2

The presented dataset is based on one of the several methods/models that can be used to track water from evapotranspiration until it contributes to precipitation. I believe it is necessary to fully discuss the assumptions and limitations of this model, and anticipate how the results from WAM-2layer can be different from other models. Indeed, the authors have made such effort by comparing with QIBT estimates. However, it is important to discuss the possible biases, especially with respect to more sophisticated methods like water vapor tracers embedded in climate models, which tend to fully resolve the physical processes that moisture can possibly be involved in the climate models. Such discussion can be critical for the future users by providing caveats and advantages of this dataset and also how they can anticipate the differences with other methods and to be aware of when and where such differences will mostly likely occur.

A2.2) Reply to comment 2

Indeed, the possible differences with respect to other models are generally of importance. We agree that water vapor tracers embedded in climate models are a more sophisticated way of tracking, but climate models are generally not as good as reanalysis products in describing the current climate and they carry no information of the actual source-receptor relations for historical dates as they only reflect the climate over a longer time span (which means that a historical drought, e.g. summer 2018 in Europe, does not occur in summer 2018 in a climate model). Moreover, we believe that a full model comparison would require a different study setup (including a comparison to other models and other datasets, which would require the involvement of the larger moisture tracking community) and is beyond the scope of this paper. This would be a paper on its own. An example for a model comparison is the following paper which compares WAM, 3D water tracking and the most complex RCM tag method:

-Van der Ent, R. J., Tuinenburg, O. A., Knoche, H. R., Kunstmann, H. and Savenije, H. H. G.: Should we use a simple or complex model for moisture recycling and atmospheric moisture tracking?, Hydrol. Earth Syst. Sci., 17(12), 4869–4884,

doi:10.5194/hess-17-4869-2013, 2013.

It was shown in this paper that the current WAM-2layers method does quite well at simulating similar results to the online tracking (RCM tag) method.

An alternative would be to discuss all these points without the solid basis of a suitable study setup. This, on the other side, would open a discussion paragraph which would be too speculative in our opinion.

A2.3) Resulting changes within the manuscript

Based on the provided arguments we would propose to stick to a version without discussion on a comparison between different models.

B) Author's reply with regard to the comments of anonymous referee #2

B1.1) General comment 1

The dataset was built using ERA-Interim data as input. Even when ERA-Interim had one of the best representations of different aspects of the hydrological cycle (now improved in ERA5), it also had some biases in representing variables like evaporation and precipitation. It would be helpful if the authors discuss the potential implications of these biases on the accuracy/applicability of their new dataset. For example, the authors could add maps of biases of evaporation and/or precipitation, at least for some regions. This would give the reader an idea about where the new dataset could have the largest uncertainties. A discussion about the biases in the input evaporation and precipitation could also help on the comparison with the QIBT estimates.

B1.2) Reply to general comment 1

Thanks for bringing this point up. We agree with you that Era-Int had some biases in representing variables like evaporation and precipitation and that we could discuss this a bit more in detail.

B1.3) Resulting changes within the manuscript

As a result, we are working on a new subchapter 4.2 as shown below and attached as well plots for the global annual evaporation and precipitation within the SI which could be used by the reader for individual comparisons.

-4.1 Possible uses of the dataset

-4.2 Critical reflections on the used input data

-4.3 Comparison to other datasets

The new subchapter discusses the occurrence of possible biases on continental scales and uses a comparison to a publication which refers to the observed state of the global water cycle over continents:

-Rodell, M., Beaudoing, H. K., L'Ecuyer, T. S., Olson, W. S., Famiglietti, J. S., Houser, P. R., Adler, R., Bosilovich, M. G., Clayson, C. A., Chambers, D., Clark, E., Fetzer, E. J., Gao, X., Gu, G., Hilburn, K., Huffman, G. J., Lettenmaier, D. P., Liu, W. T., Robertson, F. R., Schlosser, C. A., Sheffield, J. and Wood, E. F.: The observed state of the water cycle in the early twenty-first century, J. Clim., 28(21), 8289–8318, doi:10.1175/JCLI-D-14-00555.1, 2015.

In addition to the comparison with presented paper, we intend as well to include a comparison to the recently published ERA5 reanalysis.

B2.1) General comment 2

The current presentation of the comparison with estimates from the QIBT leaves the impression that there is little in common between both datasets, and no indication as to which dataset could be closer to a ground-truth. Given the accumulation of uncertainties, due to input reanalysis data and details of the water tracking methods, it is no surprise to have differences. However, not only there is kind of a systematic difference (where QIBT yields larger values than WAM-2layers, as discussed by the authors), but not even the rankings of the countries coincide between tables. Maybe, in order to look for information in both datasets that could be robust to the differences on input

data and water tracking method, the authors could include an example of the ranking of sources for a given country, and check the consistency or lack of consistency (now in terms of ranks, not original fractions) between both datasets. This would provide the reader with a better sense on what information is the most robust in the new dataset.

B2.2) Reply to general comment 2

Thank you a lot for this comment. With regard to the high differences to the estimates derived through the QIBT method, we assume that most differences are stemming from different input data rather than the tracking method itself. This is because WAM-2layers was found to reach similar results to Lagrangian models:

-Van der Ent, R. J., Tuinenburg, O. A., Knoche, H. R., Kunstmann, H. and Savenije, H. H. G.: Should we use a simple or complex model for moisture recycling and atmospheric moisture tracking?, Hydrol. Earth Syst. Sci., 17(12), 4869–4884, doi:10.5194/hess-17-4869-2013, 2013.

-Van der Ent, R. J. and Tuinenburg, O. A.: The residence time of water in the atmosphere revisited, Hydrol. Earth Syst. Sci., 21(2), 779–790, doi:10.5194/hess-21-779-2017, 2017.

However, as a general trend we observed indeed higher overlaps with regard to the rankings when we consider the sources of precipitation for specific countries.

B2.3) Resulting changes within the manuscript

As a resulting change, we highlighted within the manuscript the point that most differences might be stemmed from different input data. Furthermore, we stress now that for individual countries overlaps between the two datasets might be larger. This is as well exemplified within a new section of the supporting information which contains for the three example countries of the paper (Brazil, Egypt and Laos) an additional comparison of the top ten contributors to precipitation between the datasets. This comparison can be found within the attached supplement file (Table S3 to S5.pdf). In the following,

the text amendments within the manuscript are shown:

->"Differences regarding the tracking method itself, on the other hand, might probably play a less important role as WAM-2layers was found to reach generally similar results to Lagrangian models (Van der Ent et al., 2013; Van der Ent and Tuinenburg, 2017)."

->"Larger overlaps between the two datasets could partly be identified while focusing on the top contributors for precipitation over individual countries. This is exemplified within Table S3 to S5 of the supporting information which provide an overview on the top ten sources of precipitation for the sample countries Brazil, Egypt and Laos in comparison to the 3D QIBT method. Especially the country Laos shows in this context a relatively high match regarding the appearance of sources and their ranking to each other."

B3.1) Technical comment a

In Table 1, please add description of the "25680" and "8684" values, to help the reader to more easily understand the type of sources and receptors included in this table, without need to refer to distant parts of the paper (these specific values are described well above (Lines 104-108), and then well below (Table 2)).

B3.2) Reply to general comment a

Thank you, for clarifying that more description at this point would be useful.

B3.3) Resulting changes within the manuscript

We modified the heading of Table 1 as followed:

-Table 1 Exemplary source-receptor (evaporation-precipitation) matrix – source cells refer to considered land cells only whereas receptor cells cover all grid cells between 79.5° N and 79.5° S latitude

B4.1) Technical comment b

I tried the link for the visualization of the evaporationsheds (http://wftools.see.tu-berlin.de/wf-tools/evaporationshed/#/), but it did not work (on April 26, 2020). Please check.

B4.2) Reply to technical comment b

In the original discussion paper the link shows an additional hyphen as highlighted below: http://wf-tools.see.tu-berlin.de/wf-tools/evaporationshed/#/ Thus, the occurred problem was perhaps just related to a small typo.

B4.3) Resulting changes within the manuscript

At this point, there is no need for changes.

B5.1) Technical comment c

Individual files (e.g. http://hs.pangaea.de/model/WAM-2layers/Link-etal_2019/Interannual/2018.zip) as in "LinkA-etal_2019_inter-annual.tab" are very large (19GB). It would be helpful to have smaller examples also available for download, for example for just one country or basin, in order to test the rest of tools available with this dataset (as in http://hs.pangaea.de/model/WAM-2layers/Link-etal_2019/readme.pdf). I think that having the possibility for this kind of simple tests would help the readers and encourage the potential users to actually download and work with this dataset.

B5.2) Reply to technical comment c

Thank you for this last technical comment. Indeed, a provision of download links for the inter-annual data referring to countries or basins would facilitate in certain cases the download. However, the delineations for the regions of interest might differ from case to case and with 265 countries and 8223 basins this would show on the other side the risk of a less clear arrangement / structure of the dataset. At this point, we had to find a kind of compromise with the platform operator PANGAEA. The recommended way was, to split the very large inter-annual dataset which showed around 388 GB into 18

yearly datasets with around 19 GB and to have one basic dataset for the averages over the whole period. We assume that researchers who want to work more in detail with this data might handle this data volume.

However, with regard to the average fate of land evaporation, the screening tool (http://wf-tools.see.tu-berlin.de/wf-tools/evaporationshed/#/) allows in addition to do some quick testing on a land grid cell basis without the need to download larger files.

B5.3) Resulting changes within the manuscript

Based on the argumentation above we would propose to stick to the actual data storage if possible.

Please also note the supplement to this comment:
https://www.earth-syst-sci-data-discuss.net/essd-2019-246/essd-2019-246-AC1-supplement.pdf

―――――――――――――――――――――――

**Supplement:**

**Table S1 Top 10 sources of precipitation for the country Brazil – Comparison between results from the 3D QIBT model (fed with NCEP-DOE AMIP-II data (Kanamitsu et al., 2002) for wind and evaporation and CMAP data (Xie and Arkin, 1997) for precipitation) and the WAM-2layers model (fed with ERA-Interim data (Berrisford et al., 2011; Dee et al., 2011)) – Sources appearing in both lists are displayed in bold font; UA = Unassigned**

| Rank | Top 10 sources of precipitation for Brazil in % | | | |
|------|------------------|------|------------------|------|
| | 3D QIBT | | WAM-2layers | |
| 1 | **Brazil** | 46.3 | **Sea** | 63.3 |
| 2 | **Sea** | 43.3 | **Brazil** | 28.9 |
| 3 | **Bolivia** | 2.4 | **Bolivia** | 1.2 |
| 4 | **Peru** | 1.3 | **Peru** | 0.6 |
| 5 | **Argentina** | 1.2 | **Argentina** | 0.6 |
| 6 | **Paraguay** | 0.9 | Angola | 0.4 |
| 7 | Nigeria | 0.6 | **Paraguay** | 0.4 |
| 8 | Côte d'Ivoire | 0.5 | Venezuela | 0.3 |
| 9 | Ghana | 0.4 | Guyana | 0.3 |
| 10 | Other land | 2.9 | Other land + UA | 4.0 |

**Table S2 Top 10 sources of precipitation for the country Egypt – Comparison between results from the 3D QIBT model (fed with NCEP-DOE AMIP-II data (Kanamitsu et al., 2002) for wind and evaporation and CMAP data (Xie and Arkin, 1997) for precipitation) and the WAM-2layers model (fed with ERA-Interim data (Berrisford et al., 2011; Dee et al., 2011)) – Sources appearing in both lists are displayed in bold font; UA = Unassigned**

| Rank | Top 10 sources of precipitation for Egypt in % | | | |
|------|------------------|------|------------------|------|
| | 3D QIBT | | WAM-2layers | |
| 1 | **Sea** | 82.4 | **Sea** | 76.6 |
| 2 | **Libya** | 3.3 | **Egypt** | 2.7 |
| 3 | **Egypt** | 2.8 | **Turkey** | 1.9 |
| 4 | **Algeria** | 1.4 | **Greece** | 1.2 |
| 5 | **Greece** | 1.0 | **Libya** | 1.1 |
| 6 | Spain | 1.0 | **Sudan / South Sudan** | 0.9 |
| 7 | **Sudan / South Sudan** | 0.7 | **Algeria** | 0.9 |
| 8 | Morocco | 0.6 | Nigeria | 0.8 |
| 9 | **Turkey** | 0.6 | United States | 0.8 |
| 10 | Other land | 6.0 | Other land + UA | 13.1 |

**Table S3 Top 10 sources of precipitation for the country Laos – Comparison between results from the 3D QIBT model (fed with NCEP-DOE AMIP-II data (Kanamitsu et al., 2002) for wind and evaporation and CMAP data (Xie and Arkin, 1997) for precipitation) and the WAM-2layers model (fed with ERA-Interim data (Berrisford et al., 2011; Dee et al., 2011)) – Sources appearing in both lists are displayed in bold font; UA = Unassigned fractions due to system boundary losses**

| Rank | Top 10 sources of precipitation for Laos in % | | | |
|---|---|---|---|---|
| | 3D QIBT | | WAM-2layers | |
| 1 | **Sea** | 56.3 | **Sea** | 70.0 |
| 2 | **Thailand** | 9.4 | **Thailand** | 6.4 |
| 3 | **Burma** | 6.9 | **Laos** | 4.1 |
| 4 | **Laos** | 6.6 | **India** | 3.9 |
| 5 | **India** | 6.2 | **Burma** | 3.6 |
| 6 | **China** | 5.2 | **China** | 3.4 |
| 7 | **Vietnam** | 4.0 | **Vietnam** | 1.9 |
| 8 | **Cambodia** | 2.2 | **Cambodia** | 1.3 |
| 9 | Pakistan | 0.4 | Indonesia | 0.4 |
| 10 | Other land | 2.9 | Other land + UA | 5.0 |